# An Improved Kinect Recognition Method for Identifying Unsafe Behaviors of Metro Passengers

**DOI:** 10.3390/s22197386

**Published:** 2022-09-28

**Authors:** Ying Lu, Yifan Yu, Xifan Lv, Tingting Tao

**Affiliations:** 1School of Resources and Environmental Engineering, Wuhan University of Science and Technology, Wuhan 430081, China; 2Hubei Industrial Safety Engineering Technology Research Center, Wuhan University of Science and Technology, Wuhan 430081, China; 3Safety Business Department, Sinosteel Wuhan Institute of Safety and Environmental Protection, Wuhan 430081, China

**Keywords:** unsafe behavior recognition, eigenvectors, joint angles, dynamic time warping (DTW), metro passenger

## Abstract

In order to solve the problem of the low action recognition accuracy of passengers’ unsafe behaviors caused by redundant joints, this study proposes an efficient recognition method based on a Kinect sensor. The method uses the pelvis as the starting point of the vector and high-frequency bone joints as the end point to construct the recognition feature vector. The joint angle difference between actions is obtained by using the cosine law, and the initial test result is converted into action similarity combined with the DTW similarity algorithm. Considering the combination of 3 angle features and 4 joint feature selection methods, 12 combined recognition models are formed. A comparative experiment was carried out to identify five types of unsafe behaviors of metro passengers—punch, one-armed call for help, arms for help, trip forward and trip backwards. The results show that the overall selection of joints has a poor similarity effect and cannot achieve the purpose of recognition. The overall recognition model effect of the local “pelvis divergence method” is higher than that of the local “adjacent joint method”, and the local “pelvis divergence method” has the highest recognition result of the maximum angle difference model, and the recognition results of the five unsafe behaviors are 86.9%, 89.2%, 85.5%, 86.7%, and 88.3%, respectively, and the recognition accuracy of this method is 95.7%, indicating the feasibility of the model. The recognition results are more concentrated and more stable, which significantly improves the recognition rate of metro passengers’ unsafe behavior.

## 1. Introduction

When a large number of metros are put into use in various cities, the frequency of metro operation interruptions or passenger casualties also increases [1]. The economic losses and casualties caused by the frequent occurrence of metro accidents not only affect the economic benefits of enterprises, but also destroy social stability to a certain extent. Human factors are the most important reasons for metro accidents. Among the 128 operational cases collected by Wanxin et al. [2], 85 accidents were caused by metro passenger behavior, accounting for 66% of the total number of accidents.

In order to strengthen the management of metro passenger behavior, video surveillance devices have been installed in densely populated areas such as metro entrances, turnstiles, and escalators. Traditional detection and recognition mainly rely on monocular vision [3]. However, these devices have problems such as low recognition accuracy and poor stability, and are not able to identify unsafe behaviors of passengers [4,5], and there is no abnormal behavioral warning function. Staff need to take turns on duty to supervise the activities of passengers in the station. After an accident, they can only collect evidence by watching the video, which cannot be dealt with in time [6]. The emergence of a series of depth camera sensors, such as Kinect launched by Microsoft, has important theoretical significance and practical application value for research on intelligent recognition of metro passengers’ unsafe behavior.

Behavior recognition is performed through the detection and tracking of acquired video sequences, using the related computer vision, image processing and other technologies to describe specific actions and realize behavior recognition. The accuracy of recognition results depends on the extraction of behavior features and the selection of appropriate recognition methods [7,8,9,10,11,12,13,14,15,16]. In order to accurately describe the behavior features for subsequent action recognition, different action features are extracted according to the types of the acquired basic data information. The commonly used feature parameters of different recognition systems include static feature parameters and dynamic feature parameters. Typical unsafe behavior recognition systems and their characteristic parameters are compared and analyzed, as shown in Table 1.

As shown in Table 1, the behavior recognition technology based on video information and image information is easily disturbed by external environmental factors such as light and object occlusion, and requires preprocessing of the original data, which increases the overall recognition difficulty, complicates the implementation process, and has low repetition rate. Kinect bone information recognition based on infrared structured light detection can effectively avoid the interference of environmental factors [17]. Considering the complexity of the metro environment, the use of Kinect sensors to identify unsafe behaviors of metro passengers is based on skeletal information, which is more suitable than using video methods to obtain data.

However, since the identification of metro passengers’ unsafe behavior needs to be performed in real time and requires accuracy, when using Kinect for behavior recognition, algorithms have a great impact on the recognition effect. Therefore, according to the characteristics of metro passenger behavior, selecting an appropriate algorithm is a topic that needs further discussion.

The current Kinect-based behavior recognition algorithms can mainly be divided into two categories, one is the state space method and the other is the template-matching method. The state space method includes the hidden Markov model [18], a Bayesian network [19], a BP neural network [20] and a transfer learning model [21,22]. These methods can be easily used to obtain data, and only need traditional video information, picture information, etc., to perform subsequent behavior recognition. However, this method requires a large number of training data to learn model parameters, usually combining machine learning and deep learning methods, and the determination of observed values of model parameters often has certain errors; sometimes for the convenience of model solving, it will affect the generation process of real data [23,24].

The other approach is the template-matching method that uses frame-to-frame matching [25], frame fusion matching [26] and key frame matching [27], among which the dynamic time warping (DTW) algorithm is the most typical one. This method matches the behavior feature sequence in the sample with the corresponding standard behavior, and then determines the distance between the sequences to obtain the action similarity. Its advantage is that it does not need a large number of samples as the recognition basis, and the algorithm process is simple and easy to operate [28]. Therefore, considering that passengers’ unsafe behaviors are mostly simple actions rather than complex continuous actions, using the DTW algorithm can detect the behaviors more quickly and easily, which meets the requirement of metro operation for speed-up accident prevention and rapid emergency response.

In the field of DTW, previous studies mostly focus on improving the accuracy of behavior recognition by replacing the Euclidean distance with the Mahalanobis distance, the coupled hidden Markov model, etc. [29,30,31], but few studies consider the method of building bone feature vectors and the selection of DTW model parameters on the recognition effect.

In regard to the building bone feature vectors, most of the action features of bone information use the vectors formed between adjacent joint points as the extracted action feature vectors. For example, YU Ruiyun [32], LU Zhongqiu [33], etc., select the joint coordinates of the main parts of the body, and then calculate the adjacent bone joint point vectors to obtain and identify the similarity between standard actions and test actions. However, this joint selection method has the disadvantage of redundant joints, and is also affected by factors such as the height and body of the recognized object and the body offset during the recognition process, resulting in the low robustness of the recognition results.

As for the selection of DTW model parameters, most of the current studies [34,35] determine the joint angle feature by calculating the mean of the angle set, but the DTW algorithm finds the optimal road strength, and only calculating the mean of the angle set may not achieve the optimal recognition effect. In other words, which angle difference is most conducive to the recognition result is unknown. Furthermore, the influence of coupling between bone feature vectors and angle difference parameters on the recognition results needs further study.

To sum up, there are two problems to be solved. (1) In view of the low similarity of motion recognition using the overall adjacent joint method, and the problem that individual unsafe behaviors are mistakenly recognized as safe behaviors, resulting in reduced recognition accuracy, this study adopts the local pelvic divergence method, hoping to improve the similarity of motion recognition and thus improve the accuracy of behavior recognition. (2) It is of great significance to conduct a comparative study on the mean value, the maximum value of the angle and the angle sum to obtain the optimal parameter values.

In this study, based on the skeletal information obtained by Kinect, the overall adjacent joint method, the overall pelvis divergence method, the local adjacent joint method and the local pelvis divergence method were used for comparative analysis, and the cosine method was used to obtain the joint angle difference between actions. The similarity calculation model of the action feature quantity is established by using the angle difference, and the optimal path between the actions is obtained by using the DTW algorithm and converted into the action similarity. Then, the unsafe behavior recognition experiment of metro passengers is carried out, and the similarity score of each action is obtained. Comparing and analyzing the experimental results, the optimal recognition model is obtained. Finally, with the help of the MATLAB App Designer interface development platform, the identification and early warning system of unsafe behavior of metro passengers is designed to verify the recognition accuracy of the optimal recognition model. Additionally, this method is compared with traditional identification methods and similar studies by other scholars to verify the scientific validity of the identification model.

The main contributions of this paper are as follows. (1) Through the comparative analysis of the metro passengers’ unsafe behavior identification experiments, an optimal identification method is determined, the feasibility and accuracy of the method are verified, and a new technical means for the management and control of metro passengers’ unsafe behavior is provided. (2) The improved Kinect recognition method is applied to the metro station to solve the problem that the recognition effect of passengers’ unsafe behaviors is not ideal in the complex environment of the metro station.

## 2. Materials and Methods

### 2.1. The DTW Algorithm Based on Unsafe Behavior of Metro Passengers

#### 2.1.1. Unsafe Behavior Identification

Among the causes of metro operation accidents, the behavior of passengers is the most complex and changeable [36], but there are relatively few studies on unsafe behaviors in their usual state. Therefore, the author selects the identification objects based on the existing literature, metro accident cases and related behavior monitoring systems. According to GB/T-2007 “metro Operation Safety Evaluation Standard” in China, eight behaviors that violate the normative standards are pointed out in the code of conduct for passengers, such as carrying dangerous goods, riding the elevator in the wrong direction, jumping from the platform, fighting and another four behaviors. Among the 91 cases of passenger unsafe behavior accidents that have been counted by the author, falls and fights account for 68%, of which falls account for 45% and fights account for 23%. Fall behavior is refined, and can be divided into forward falls and backward slips. In addition, in the special report “Frontier Application of Artificial Intelligence in the Field of Public Security Video Big Data Analysis”, it is pointed out that the behavior of calling for help is among the behaviors of public security video surveillance. The behavior of calling for help can be divided into one-arm calling for help and two-arm calling for help. Therefore, this kind of behavior can also be regarded as the unsafe behavior of metro passengers, so that the objects of metro video surveillance can be diversified and standardized. For ease of expression, this article still uses unsafe behavior as the general term for identification objects, which specifically refers to unsafe behavior or abnormal behavior of metro passengers.

Accordingly, combined with the characteristics of Kinect equipment, according to the above survey results and related literature [37,38,39,40], five unsafe behaviors—punching (fighting and fighting), one-armed call for help, two-armed call for help, falling forward and slipping backward—were selected as the action recognition objects.

#### 2.1.2. Bone Information Acquisition

This research uses the latest generation of Azure Kinect released in 2019, through the application programming interface provided by the Kinect Body tracking SDK (software development kit) development kit and the Kinect SDK to collect 32 joint points at a frequency of 30 f/s, providing that at each joint, the three-dimensional coordinates, azimuth, confidence level and other information of the skeleton can be obtained by using the bone tracking technology to obtain the joint point positions of the standard action and the test action, as shown in Table 2. Compared with Kinect V2, the device can detect 7 more joint points of nose, left eye, right eye, left ear, right ear, left clavicle and right clavicle. The 32 joints captured by Azure Kinect are shown in Figure 1.

### 2.2. Improved Kinect Recognition Method

Each frame of Azure Kinect will generate coordinate data of 32 joint points, that is, output the three-dimensional coordinates (x, y, z) of each joint point, and the final action output result will be 32 × 3 = 96 dimensions. This will lead to the redundancy of the data model, and it is also inconvenient for the operation and analysis of the data. Therefore, it is necessary to exclude some joints that have little correlation with action features, such as nose, eyes, ears and other joints. In addition, as can be seen from Figure 1, the positions of some nodes are relatively concentrated, such as hand joints and wrist joints (left and right), foot joints and ankle joints (left and right). If they are directly used as features, it will not only increase a certain amount of calculation, but also reduce the effect of action recognition to a certain extent. Therefore, the joint points of similar parts should be selected to avoid too similar and repeated joint parts. In view of the above situation, the characteristics of five metro passenger unsafe behaviors are analyzed from the two aspects of the overall action and the action part, and the joint points with obvious action characteristics are selected. In order to facilitate the description of the subsequent action model, the joint points involved in the action are coded.

#### 2.2.1. Overall Joint Point Selection

In order to describe different actions with the same feature vector, several sets of vectors that can describe the behavior of the upper body and lower body of the human body are usually selected, respectively. For example, in order to ensure the integrity of the movements, Li Hongbo et al. [41] extracted 14 active joint points that constitute the left and right arms, upper torso, and left and right lower limbs. Therefore, according to the above selection method, considering the overall situation of the action, a group of skeleton points suitable for 5 kinds of unsafe behaviors of metro passengers is determined. The specific selection method is shown in Figure 2, and the selected joint points are coded.

#### 2.2.2. Local Joint Point Selection

In order to describe different actions more accurately, a common approach normally includes (1) analyzing the unique action characteristics of each action, (2) treating each action as an independent analysis individual, (3) analyzing the high-frequency active joint points of each action one by one, and (4) further selecting the designated bone joints for each action point. For example, Ding Weili [42] and others only consider hand joints, elbow joints and head joints for subsequent identification when identifying movements without looking into lower limb movements. By analyzing the action characteristics of five kinds of unsafe behaviors of metro passengers, the joint points of the corresponding actions are determined, and the joint points selected for the overall action and the action part are sorted, as shown in Table 3.

#### 2.2.3. Joint Vector Selection

In terms of joint feature vector selection, the traditional joint feature vector is usually a vector formed by connecting adjacent bone joint points. For example, the vector formed by the right shoulder A and the right elbow B is used as the feature vector, and this traditional selection method is named the “adjacent joint method”.

However, this method is easily affected by factors such as the different body shapes of the action objects and the body offset during the recognition process, which may reduce the similarity of the recognition results. Accordingly, a method for selecting joint feature vectors is proposed, which takes the hip joint (pelvis) as the starting point O of the vector and the specific bone joint point as the end point of the vector. For example, starting from the point O of the hip joint and pointing to the joint point A of the right shoulder, the formed vector is the model feature vector, which is named the “pelvis divergence method”, and improve the similarity of the recognition results.

Considering the two situations of the overall action and the action part, comparing the above two vector selection methods, four methods for selecting the skeleton space feature vector are obtained. Taking the action of punching as an example, the specific selection process of the four methods is described, respectively.

Overall “adjacent joint method”: The changes of joint movements of the whole body are considered as a whole, so as to represent the movement shape, and the upper and lower limbs are selected as the characteristic joint vectors. The vector is constructed according to the selection method of the “adjacent joint method” and the action joint points determined in Table 3, as shown in Figure 3.

The overall “pelvis divergence method”: According to the selection characteristics of this method, the hip joint is used as the starting point, and the joint points corresponding to the upper and lower limbs are used as the vector end points to construct feature vectors, as shown in Figure 4.

Local “adjacent joint method”: According to the analysis of the characteristics of the punching action and the determination of joint points, three points A–C are selected, and the joint vectors formed by the three points are connected. The constructed feature vector is shown in Figure 5.

Local “pelvis divergence method”: Take the right-hand punching action as an example, the point O of the hip joint is the starting point of the vector, and the points A–C are the end points of the vector. The specific selection method is shown in Figure 6.

To sum up, according to the four feature vector selection methods, combined with the skeleton joint points determined in Table 3, the selection of skeleton space feature vectors for five types of metro passengers’ unsafe behaviors is summarized; see Table 4 and Table 5 for details.

#### 2.2.4. Joint Angle Extraction

Taking the “pelvis divergence method” as an example, the degree of movement difference is determined by selecting the angle difference formed by the same joints in the standard action and the test action as the feature quantity, as shown in Figure 7. The higher the degree of similarity of actions, the smaller the angle difference formed, and the specific value of the angle can be obtained from the cosine theorem:(1)$$\theta =arc\cos\frac{\vec{AB}\cdot \vec{ab}}{\left|\vec{AD}\left|\cdot \right|\vec{ab}\right|},$$

Among them, the standard action selection vector is $\vec{AB}$, and the test action selection vector is $\vec{ab}$. If the selected action has *n* feature vector sequences, *n* angle difference feature sequences can be obtained by the cosine theorem:(2)$$\theta_{i}=\left\{\theta_{1},\theta_{2},\theta_{3},\ldots \ldots ,\theta_{n}\right\},$$

Among them, *θ_i_* is the joint angle difference set of the action in the *i*-th frame, and *θ_n_* is the angle difference corresponding to the *n*th vector.

In order to facilitate the calculation of the subsequent DTW identification algorithm, the final *θ_i_* should be a certain value, so it is necessary to analyze the joint angle set, and at the same time, it is still necessary to ensure that the integration result can accurately reflect the difference between the standard action and the test action. Therefore, different attributes of angles are introduced, such as the mean of angle difference and the sum of angle difference. At the same time, an attribute that uses the maximum angle difference as the attribute to judge the difference of action is proposed, and the subsequent experiments are carried out to verify the recognition models of different angle attributes.

The feature quantity of the mean of angle difference is to take the mean of all the angle values obtained in the *i*-th frame as the feature quantity:(3)$$\theta_{i}=\frac{\sum^{\;}\left\{\theta_{1},\theta_{2},\theta_{3},\ldots \ldots ,\theta_{n}\right\}}{n},$$

The sum of the angle difference feature is the result of summing all the angle values obtained in the *i*-th frame as the feature:(4)$$\theta_{i}=\sum^{\;}\left\{\theta_{1},\theta_{2},\theta_{3},\ldots \ldots ,\theta_{n}\right\},$$

The maximum angle difference is to select the value with the largest angle difference from all the angle values obtained in the *i*-th frame as the feature quantity:(5)$$\theta_{i}=max\left\{\theta_{1},\theta_{2},\theta_{3},\ldots \ldots ,\theta_{n}\right\},$$

To sum up, by assigning attribute meanings to angles, three types of angle feature quantities can be obtained: the mean of angle differences, the sum of angle differences, and the maximum angle difference. After the characteristic quantity is obtained, the unsafe behavior can be identified by using the characteristic quantity combined with the DTW algorithm.

#### 2.2.5. Calculation of Action Similarity Based on the DTW Algorithm

Through the obtained standard data *S* and test data *T* two action sequences, the DTW algorithm is applied to find the best matching path between the standard action and the test action, and at the same time, the influence of time offset and deformation is minimized [43]. According to the action joint vectors selected in Table 2, the motion of each action joint vector constitutes one-dimensional time series data. Taking the punching action as an example, the data of the standard action and the test action have 3 dimensions, and the dimension of the matrix of each data set is n × 3, where n is the total number of frames of motion data.

Taking the vector shown in Figure 3 as an example, the sequence of the standard action joint vector $\vec{AB}$ is *S* = (*s_1_, s_2_, s_3_, …, s_n_*), and the sequence of the test action joint vector $\vec{AB}$ is *T* = (*t_1_, t_2_, t_3_, …, t_m_*), the elements in *S* and *T* are vectors formed by each frame in a particular coordinate system. Thus, an *m × n* cost matrix is constructed, and the (*i, j*) elements in the matrix represent the angle difference *C*(*s_i_, t_j_*) formed between *s_i_* and *t_j_*. According to the boundary, monotonicity and continuity of the DTW algorithm [44], the algorithm process is shown in Formula (6):(6)$$D\left(i,j\right)=min\left\{\begin{array}{c} D\left(i-1,j-1\right)\\ D\left(i-1,j\right)\\ D\left(i,j-1\right) \end{array}\right\}+C\left(s_{i},t_{j}\right),$$

In the formula, *i* is the abscissa of the element, 1 *< I < n*; *j* is the ordinate of the element, 1 *< j < m*; *D*(*i, j*) refers to the distance from the current position (*i, j*) and the distance that can reach the point. Sum of cumulative angles of the smallest adjacent elements.

From Formula (6), a dynamic warping path *P* with the minimum cumulative distance can be obtained, and the sum of the angle differences *C_p_(S,T)* of the joint vectors can be obtained accordingly, as shown in Formula (7):(7)$$c_{p}\left(S,T\right)=\sum_{k=1}^{i}C[s_{ik},t_{jk}],$$

In the formula, *C_p_(S,T)* is the sum of the elements of the optimal dynamic warping path in the cost matrix; *S* is the standard action sequence; *T* is the test action sequence; *l* is the step size of the path *P*.

For the multidimensional case, *s_i_* and *t_j_* are not a single bone vector, but multiple joint vectors representing the motion of the entire body. Multidimensional DTW (*S, T*) is calculated in a similar way to the one-dimensional case [45], except that *C*(*s_i_, t_j_*) needs to be redefined. The author defines *C*(*s_i_, t_j_*) as: the maximum angle difference in the multidimensional joint vector of the selected action. The output of DTW, DTW (*S, T*), is the cumulative sum of the maximum angle differences of the shortest dynamically warped path. Therefore, the smaller the cumulative sum, the higher the similarity of the two motion sequences. In order to quantify the similarity between the standard action and the test action, a similarity formula with the maximum angle difference as the feature quantity is proposed, which is represented by the percentage result; see Formula (8):(8)$$Similarityscore=1-\frac{D\;T\;W\left(S,\;T\right)}{180\times V\times l},$$

In the formula, *l* is the optimal dynamic warping path length; *V* is the number of joint vectors selected for motion evaluation.

The value range of the maximum angle difference is [0, 180°], within which the cosine function decreases monotonically, and each angle value uniquely corresponds to a cosine value. Therefore, the maximum DTW (*S, T*) of this optimal path is 180 *× V × l*.

### 2.3. Verification of the Identification Method of Metro Passengers’ Unsafe Behavior

#### 2.3.1. Experimental Purpose and Environment

In order to explore the recognition effect of the model, a comparative experiment was carried out on the recognition of the above-mentioned five kinds of unsafe behaviors of metro passengers, and the optimal model of recognition effect was obtained according to the obtained experimental results. Using three angle feature attribute parameters combined with four skeleton space feature vector selection methods, cross-combination verification is carried out, and the result is analyzed based on the similarity score of action recognition.

Experimental hardware system: one Azure Kinect sensor, host computer: Intel Core i7; 16 GB memory, NVIDIA GeForce RTX 3060 graphics card. The standard development environment is: Windows 10 operating system, Visual Studio 2019 and Matlab R2018a. The WPF (Windows Presentation Foundation) platform is used to build the experimental data acquisition interface. The interface includes modules such as image display area, action selection area, and status functional area. See Figure 8 for details.

#### 2.3.2. Experimental Process Method

There are 7 experimental participants in this experiment. All the movement data of one participant are used as the standard data, and the movement data of the other 6 people are used as the test data. Specifically define the above-mentioned punching, single/double-arm waving for help, tripping forward, and backward slipping. The action is shown in Figure 9.

Each tester stands at a distance of 2.5 m from Kinect. When the test interface shows that the device is successfully connected, click “Data Collection” to start collecting the tester’s movement data. The collection time is 10 s, and a total of 300 frames of data are collected. After the collection is completed, click “Save Data” to automatically name and save to the corresponding .xls file.6 testers performed each action 10 times, respectively, that is, there are 60 sets of test data for each action.

After saving the WPF action acquisition interface data to the specified file location, the original skeleton data are further extracted, vector transformed, DTW operation and the final similarity result through MATLAB. According to the recognition model in Section 2.2, the attributes of different angles are combined, and the specific model categories are shown in Table 6. That is, the overall “adjacent joint method”, the overall “pelvis divergence method”, the local “adjacent joint method” and the local “pelvis divergence method” and the sum of the angle differences, the average angle difference and the maximum angle difference are substituted into the DTW recognition algorithm for comparing the similarity results obtained by different models.

Finally, in order to verify the effect of the recognition method, use the application design tool of MATLAB-App designer to design an early warning system to test the accuracy of identifying unsafe behaviors. Stand and squat are added to test the recognition accuracy of the improved recognition method in this study, and compared with the traditional recognition method, which proves the progressiveness of the method in this study.

Through the identification system, the standard action and the test action are compared. According to relevant research [46], if the similarity between the standard action and the test action exceeds 75%, it is deemed to have made unsafe behavior, and the red indicator light is on. If the similarity is less than 75%, it is considered as safe behavior, and the green indicator light is on. The system interface is shown in Figure 10.

## 3. Results

### 3.1. Result of Overall Joint Selection

The recognition model results of the overall “adjacent joint” and the overall “pelvis divergence method” are shown in Figure 11. The similarity scores of each action obtained by the overall “adjacent joint method” and “pelvis divergence method” recognition models are averaged, and the specific similarity scores obtained by each group of models are shown in Table 7 and Table 8. In the overall joint recognition method, the recognition similarity of punching behavior in groups A1, B1, A2, and B2 was the lowest, which were 58.9%, 60.2%, 60.3%, and 60.3%, respectively. The similarity of the other four actions were all between 65% and 94%; however, in the C1 and C2 groups, the punching behaviors were as high as 92.1% and 95.4%, respectively, and the similarity of the remaining actions were all above 80%.

### 3.2. Result of Local Joint Selection

The recognition model results of the local “adjacent joint method” and the local “pelvis divergence method” are shown in Figure 12. Calculate the average of the similarity scores of each action obtained by the local “adjacent joint method” and “pelvis divergence method” recognition models. The specific similarity scores obtained by each group of models are shown in Table 9 and Table 10. It can be found that the similarity of the C4 group is above 85%, which is significantly higher than that of the other five groups. The similarity of punching behavior has been effectively improved in local selection.

### 3.3. Verification of Recognition Accuracy

The identification results of five unsafe behaviors and two safe behaviors by the metro passenger unsafe behavior warning system are shown in Table 11. For comparison, the test results using the traditional recognition method (using the overall adjacent joint method and the mean value of the angle difference as the feature quantities) are shown in Table 12.

## 4. Discussion

### 4.1. Comparison of the Local Recognition Model and the Overall Recognition Model

#### 4.1.1. The Shortcomings of the Overall Recognition Models

The similarity of punching behavior in groups A1, A2, B1, and B2 is low, approximately 60%. This is because the punching action can be divided into the use of force only with the hand and the force with the hand plus other parts of the body, resulting in a large difference in the action. However, the similarity of using the largest angle has been significantly improved, reaching more than 90%, and it was also found in the experiment that no matter whether the overall “adjacent joint method” or the overall “pelvis divergence method” is used, when identifying the same action or different kinds of actions (e.g., when recognizing the punching motion, the tester performed the motion of calling for help with both arms), both have the same high similarity, and the above situation occurs in different angle attribute models. For this phenomenon, the author believes that the selected action difference joints account for too little in the overall joint, which weakens the action difference, resulting in a high level of final similarity, which does not achieve the effect of distinguishing actions.

Therefore, by analyzing the experimental results, it can be seen that the identification vector is selected from the overall aspect of the action. If the selected vector has most of the relatively static joints or the movement changes are concentrated in the local joints, there will be problems of low similarity, poor effect and no distinction, which will induce the identification purpose cannot be achieved.

#### 4.1.2. The advantages of Local Recognition Models

Comparing Figure 11, Table 9 and Table 10 with Figure 10, Table 7 and Table 8, it can be found that the average similarity of the overall selection model and the local selection model in groups A and B is 76% and 73.3%, respectively—the similarity of the two is similar and is at a low level. However, in the experiment of group C, the local recognition model only considers the moving joints, which significantly improves the action difference, and can achieve a better distinguishing effect than the overall recognition model. Thus, the local recognition model is better than the overall recognition model

### 4.2. Comparison of the “Pelvis Divergence Method” and the “Adjacent Joint Method”

The similarity scores of each action obtained by the local “adjacent joint method” and “pelvis divergence method” recognition models are averaged, and the results are shown in Figure 13. Comparing and analyzing the recognition results of the local “pelvis divergence method” and the local “adjacent joint method”, only the similarity of single-arm and double-arm calls for help in group B decreased slightly, by 0.4% and 0.9%, respectively. The similarity of other actions has been significantly improved. The most improved similarity is the single-arm call for help in group A, which has increased by 15.9%. It is confirmed that the “adjacent joint method” is easily affected by factors such as the different body shapes of the action objects and the body offset during the recognition process, thereby reducing the similarity of the recognition results. Therefore, the “pelvis divergence method” can obtain higher similarity.

### 4.3. The Similarity of Using “Maximum Angle Difference” as the Feature Quantity Is the Best

According to Figure 13a, the recognition results of most actions are in descending order of group C, group B, and group A. Among them, the overall results of group A are quite different from those of groups B and C. The difference between the results of the two groups is small; according to the numerical trend, the similarity results between each action category in the group A are between 51.2% and 73.2%, and the gap is large, indicating that the stability of the action similarity is low. In contrast, the similarity of group C ranges from 75.1% to 84.5%, the recognition results for each action are generally concentrated, and the gap between the maximum and minimum values between action categories is small.

According to Figure 13b, the recognition result of group C is the highest among the three groups of models, followed by group B, and the recognition result of group A is still the lowest. The results of “method” were small, and the recognition results of the three groups of models were all higher than the local “adjacent joint method”. The recognition results of each action in group C were between 85.5% and 89.2%, the similarity was the highest, and the recognition results had good stability. Therefore, the similarity with the “maximum angle difference” as the feature quantity is the best.

Based on this, it can be seen that the similarity results of the five types of metro passenger unsafe behaviors using the maximum angle difference attribute combined with the local “pelvis divergence method” recognition model are all over 85%, which is better than other recognition models. The stability further shows that the model can better reflect the difference of action characteristics and has a good distinguishing effect.

### 4.4. The Optimal Recognition Method

It can be seen that the recognition effect of the local “pelvis divergence method” is better than that of the local “adjacent joint method”. Therefore, the local “pelvis divergence method” is used to select the feature vector, and the model with the maximum angle difference attribute as the feature quantity is the optimal parameter model.

Based on this, the similarity results of the five types of metro passenger unsafe behaviors using the maximum angle difference attribute combined with the local “pelvis divergence method” recognition model are all over 85%, which is better than other recognition models. The stability further shows that the model can better reflect the difference of action characteristics and has a good distinguishing effect.

### 4.5. Verify the Practicability of the Improved Identification Method

From the test results in Table 11, the average recognition similarities of the five unsafe behaviors are between 86.9% and 89.2%, much higher than 75% of the system warning threshold. During each experiment, when the system judged that passengers have committed unsafe behaviors, the red indicator will light on. Among the 50 test experiments of unsafe behaviors, the indicator has been on 47 times, which means the recognition accuracy of unsafe behaviors is greater than 90%. As for the three misidentification experiments, the first is that the punch was incorrectly identified as one-armed call for help, the second is that the one-armed call for help was incorrectly identified as a punch, and the third is that the arms for help was incorrectly identified as a one-armed call for help. However, all 50 red lights were on. The average recognition similarity of the two safe behaviors are 65.2% and 43.7%, much lower than 75%, which means the system judged that the passenger’s behavior was safe, and the green indicator lighted on. Therefore, in the total of 70 experiments, the system correctly identified the behavior 67 times, and erroneously identified three other unsafe behaviors, but the red indicator light was on, and the warning was still valid. The comprehensive accuracy of the improved recognition method is 95.7%.

As a comparison, the comprehensive recognition accuracy of the traditional recognition method is 72.9%. It can be seen that using the adjacent “pelvis divergence method” combined with the maximum angle difference to identify unsafe behaviors can obtain higher similarity, effectively avoiding the situation that unsafe behaviors cannot be identified due to low similarity. It shows that it is feasible to improve the recognition accuracy by improving the similarity of behavior recognition.

In order to further verify the effectiveness of the algorithm in this paper, the results of this study are compared with other related studies. Tian et al. [29] used the recognition algorithm of escalator passenger unsafe behavior based on human skeleton sequence to study escalator passenger unsafe behavior, which is similar to the study in this paper, so it is suitable for comparison. The experiment shows that the recognition accuracy of Tian’s method is 93.2%. Zhang et al. [30] used the coupled multi-hidden Markov model and depth image data to recognize human actions, and the comprehensive recognition rate was 87.16%. Gu et al. [31] used the Mahalanobis distance to improve the DTW algorithm, and the comprehensive recognition rate was 94.8%. It can be seen that the recognition method proposed in this study achieved a better expected recognition effect. Because it can perform well in complex environments, it shows strong robustness.

### 4.6. Limitations

From the recognition results in Table 9 and Table 10, it can be seen that the two methods use any angle attribute to recognize the punch action, and the similarity results are the highest, with the average results being 79% and 82.2%, respectively. The similarity result of arms for help action is low, with the average results being 65.6% and 73.3%. The author believes that there is a certain error delay in the transmission and input of data, which is a systematic error. In addition, the analysis shows that the overall low similarity of the arms for help action is mainly due to the following: if the movement speed is accelerated, such as if the double-arm waving speed is too fast, the system error will further increase. The coincidence of joint points at a certain point in time affects the stability of action information capture, resulting in deviation of joint position information. That is, the coincidence of joint points is also among the reasons for the low similarity of the action.

In addition, when comparing the accuracy, there may be deviations in the comparison results because comparison in limited to articles of similar research types, and there is no good control variable.

## 5. Conclusions

In this study, a recognition model of metro passengers’ unsafe behavior is proposed based on Kinect. In the process of building a spatial skeleton model, a new skeleton model building method and a recognition model of the DTW similarity algorithm are proposed. That is, taking the pelvis as the starting point of the vector and the high-frequency moving joint points of the action as the end point of the vector to extract the feature vector, using the maximum angle difference attribute as the feature quantity, and combining with the DTW similarity algorithm, it solves the problems of low action similarity caused by redundant joints, unclear quantification of DTW results and practical applications. In addition, the recognition score of each model is calculated by MATALB, and the group comparison experiment is carried out. The results show that the recognition effect of the overall situation is poor and the recognition purpose cannot be achieved. The effect of the local “pelvis divergence method” recognition model is generally higher than that of the local “adjacent joint method”, among which the local “pelvis divergence method” has the highest recognition results of the maximum angle difference model, and the recognition results of the five unsafe behaviors are 86.9%, 89.2%, 85.5%, 86.7%, and 88.3%, all of which are greater than 80%, indicating the feasibility of the model. Additionally, the recognition results are more concentrated and more stable, which significantly improves the recognition rate of metro passengers’ unsafe behavior.

The focus of this research is to avoid complex algorithms for effective action recognition based on a small sample size. However, during the experiment, it was found that the recognition accuracy of the double-arm waving and calling for help was generally low. In addition to the system error, the faster movement speed and the overlapping joint points were the reasons for this problem, which can be further improved and discussed in the follow-up research.

## Figures and Tables

**Figure 1 sensors-22-07386-f001:**
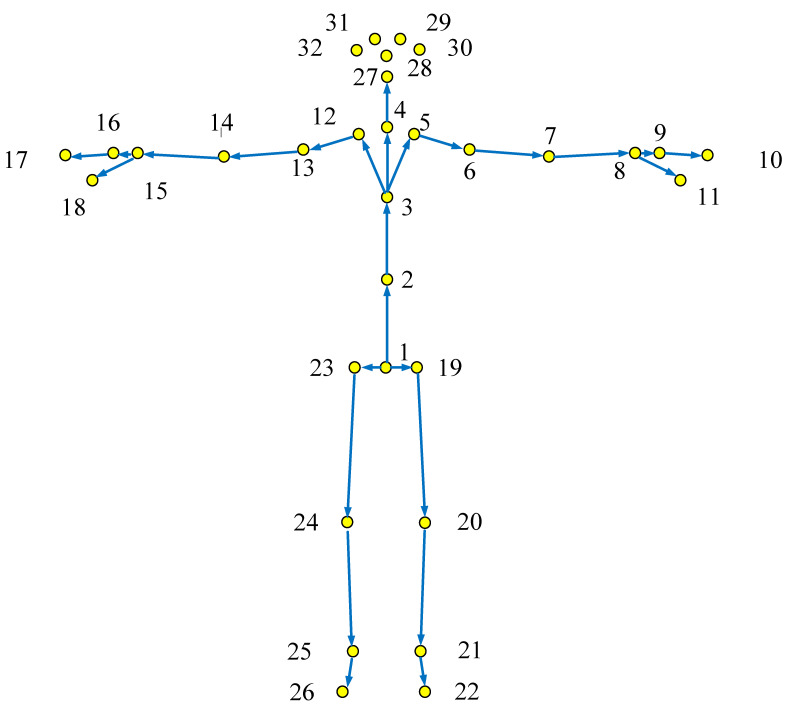
Distribution of 32 joint points.

**Figure 2 sensors-22-07386-f002:**
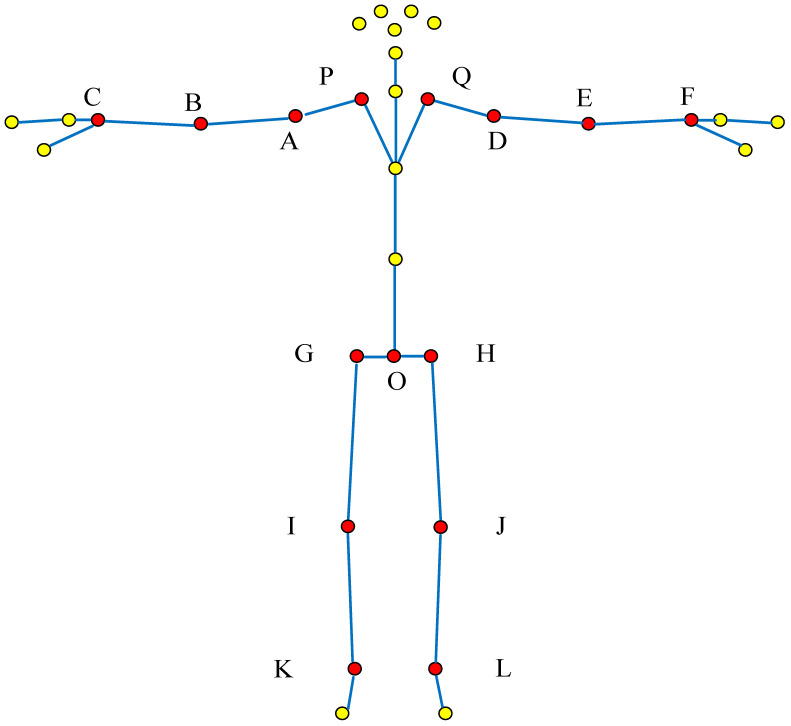
Select the overall joint point of the action (The red dot is the selected bone joint in this study, and yellow dot is bone joint point that do not need to be considered in this study. The letter is the code of selected bone joint corresponded to Table 1).

**Figure 3 sensors-22-07386-f003:**
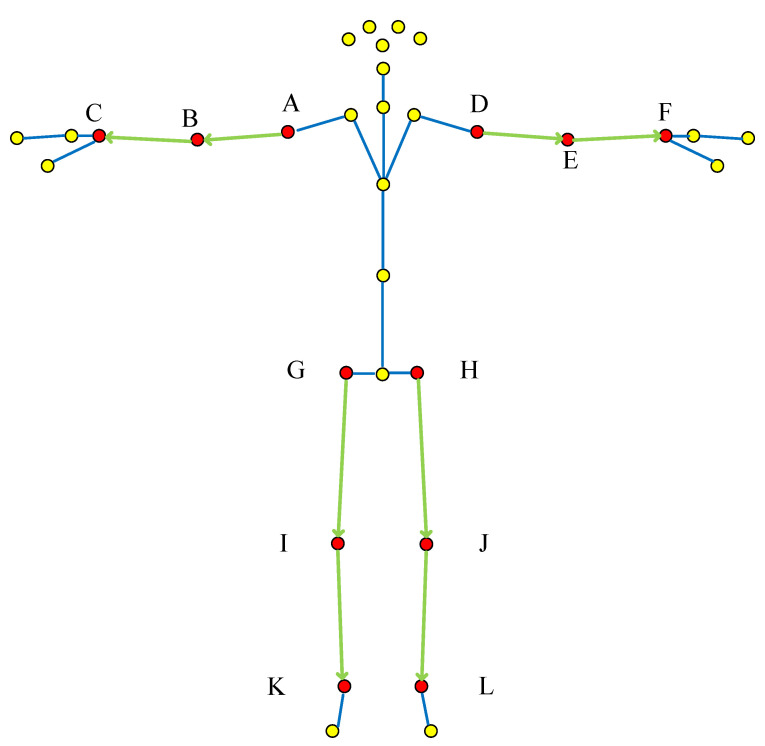
Overall “adjacent joint method” vector selection (The red dots are the selected bone joint, and yellow dots are bone joint points that do not need to be selected; Green line with arrow is the selected bone joint vector, Blue line is the connecting line of bone joints; The letters correspond to Table 1 and Figure 1).

**Figure 4 sensors-22-07386-f004:**
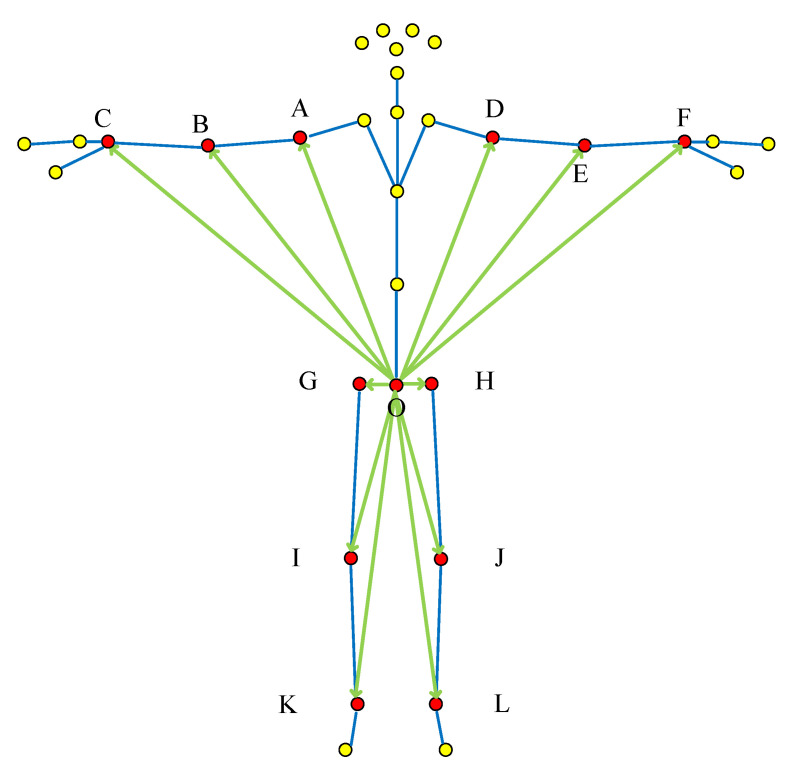
Overall “pelvis divergence method” vector selection (The red dots are the selected bone joint, and yellow dots are bone joint points that do not need to be selected. Green lines are selected joints, blue lines are not selected joints. The letters correspond to Table 1 and Figure 1).

**Figure 5 sensors-22-07386-f005:**
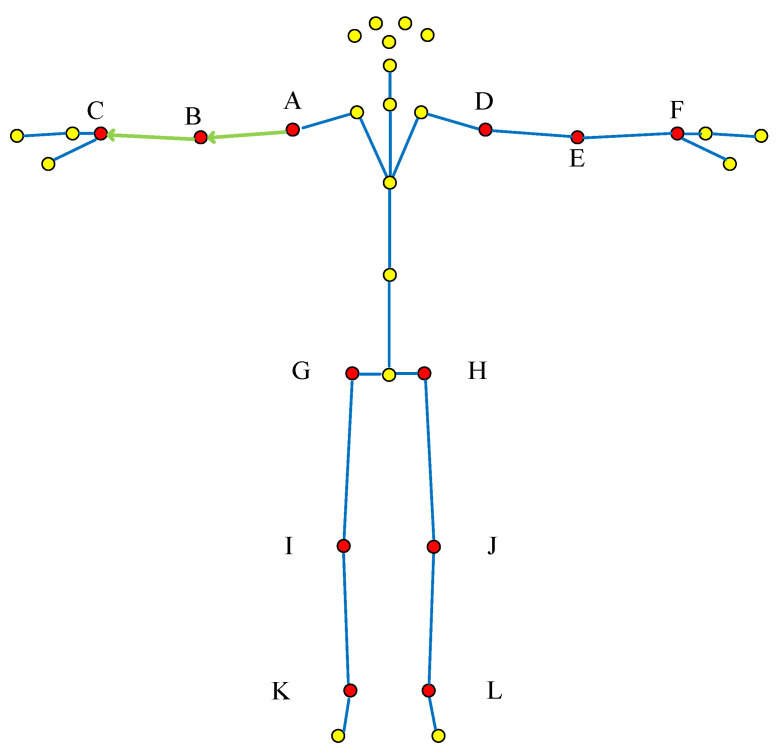
Local “joint method” vector selection (The red dots are the selected bone joint, and yellow dots are bone joint points that do not need to be selected. Green lines are selected joints, blue lines are not selected joints. The letters correspond to Table 1 and Figure 1).

**Figure 6 sensors-22-07386-f006:**
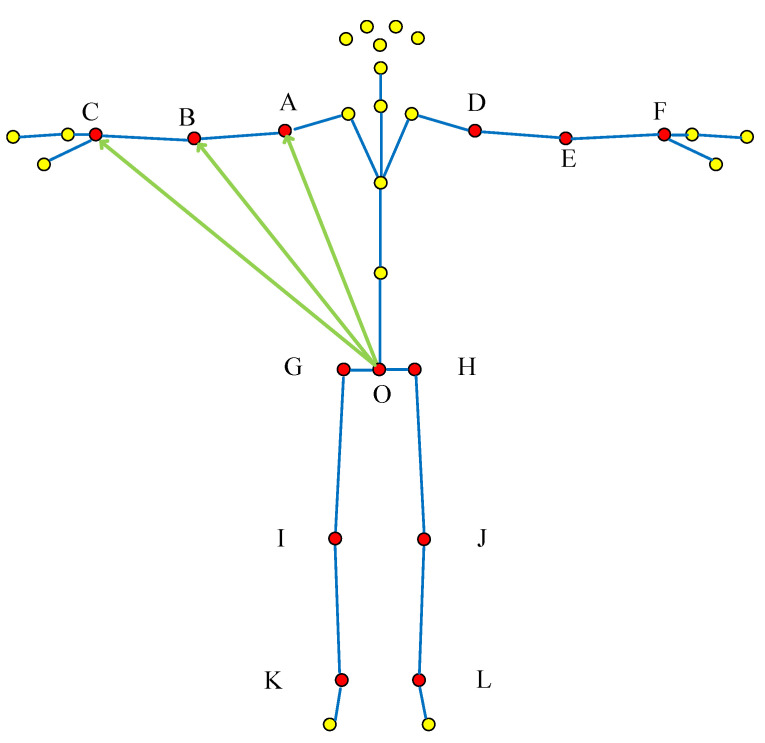
Local “pelvis divergence method” vector selection (The red dots are the selected bone joint, and yellow dots are bone joint points that do not need to be selected; Green lines are selected joints, blue lines are not selected joints; The letters correspond to Table 1 and Figure 1).

**Figure 7 sensors-22-07386-f007:**
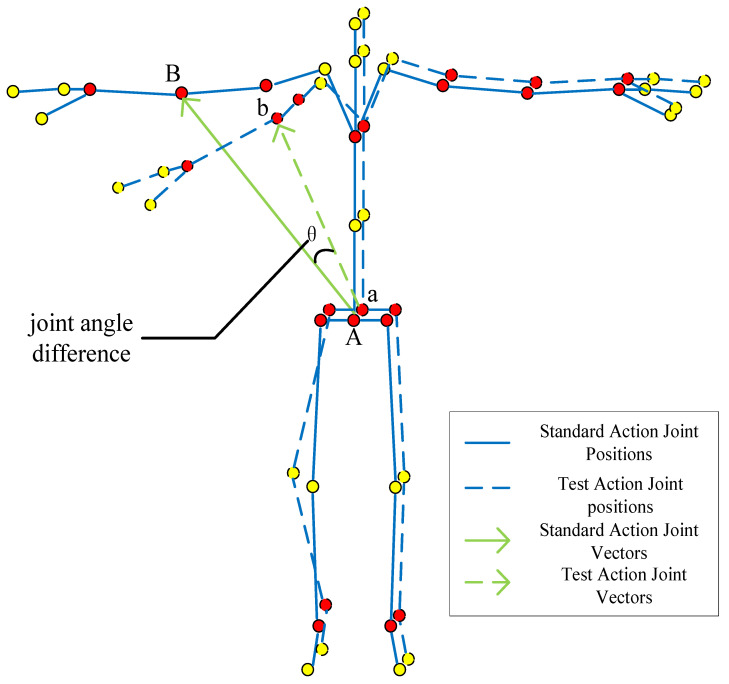
Action angle feature extraction (The red dots are the selected bone joint, and yellow dots are bone joint points that do not need to be selected; The letters correspond to Table 1 and Figure 1).

**Figure 8 sensors-22-07386-f008:**
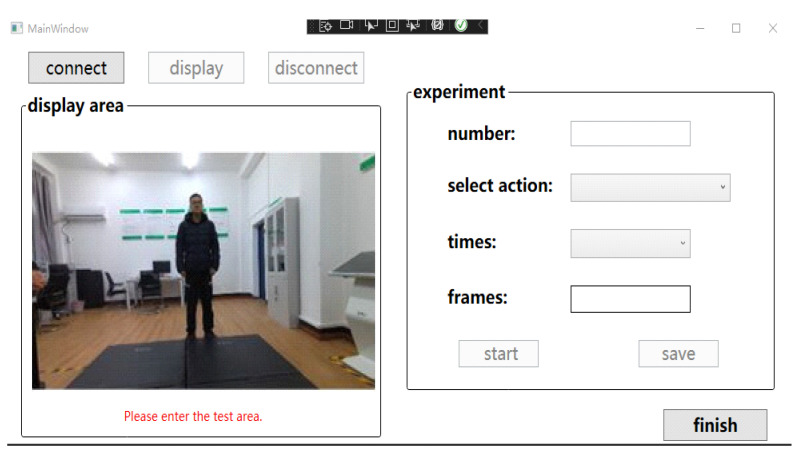
Experimental data collection interface.

**Figure 9 sensors-22-07386-f009:**
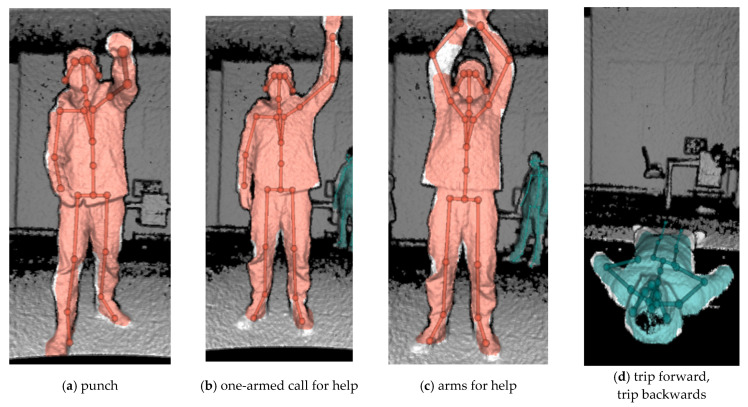
Examples of six abnormal actions.

**Figure 10 sensors-22-07386-f010:**
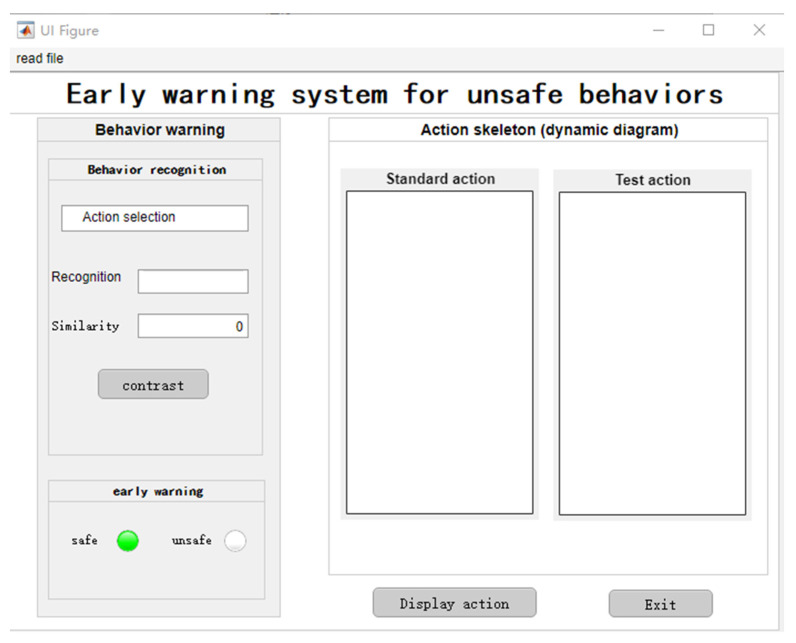
Identification and warning system of unsafe behaviors of metro passengers.

**Figure 11 sensors-22-07386-f011:**
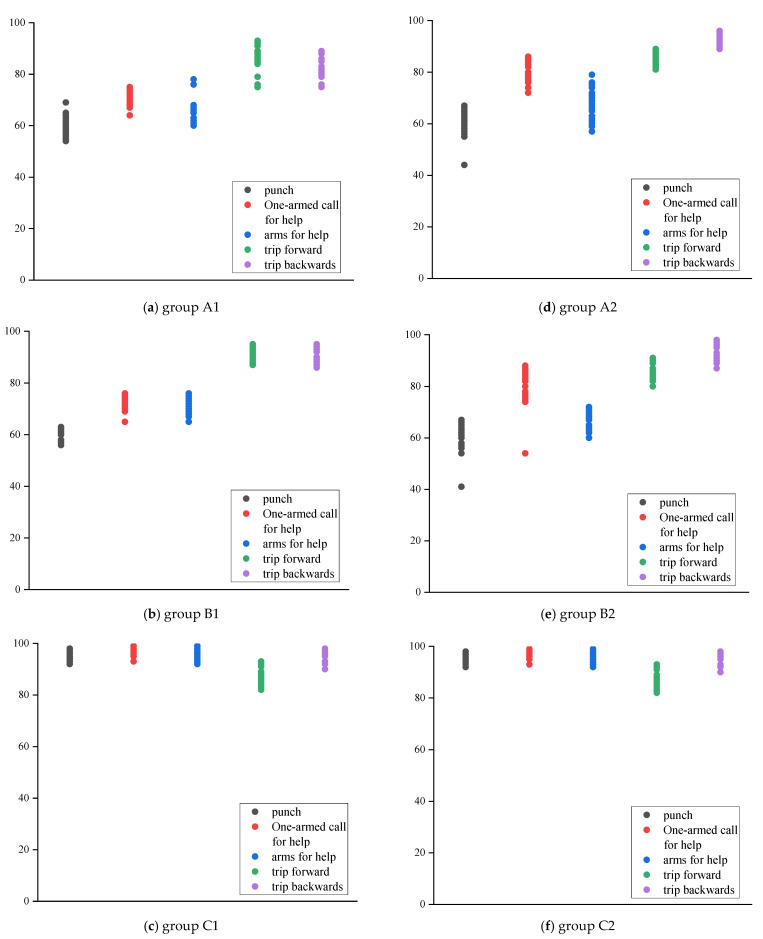
Result of overall joint selection.

**Figure 12 sensors-22-07386-f012:**
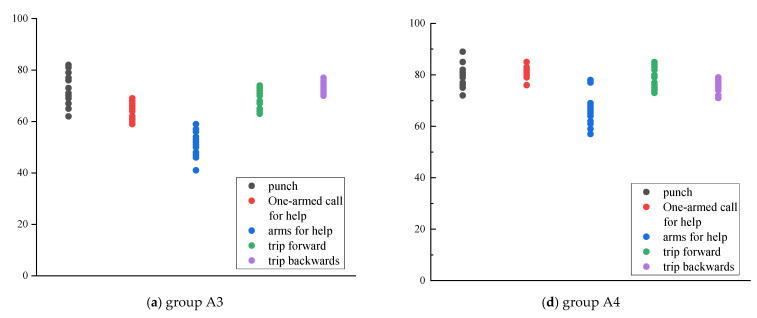
Result of local joint selection.

**Figure 13 sensors-22-07386-f013:**
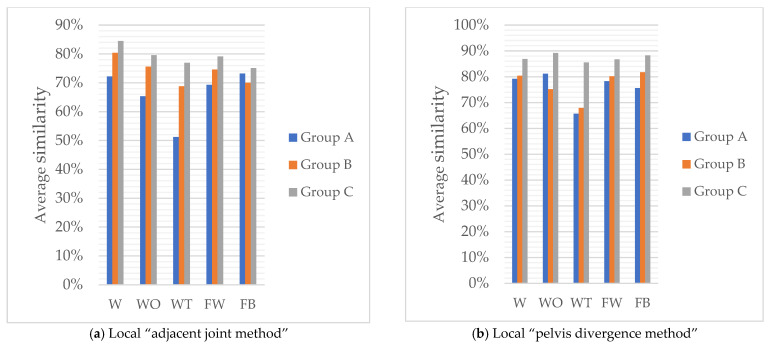
Local action recognition results.

**Table 1 sensors-22-07386-t001:** Comparison of behavior feature extraction techniques.

Behavior Characteristics	Characteristic Parameter	Type of Data	Advantages and Disadvantages of Parameters
static feature parameters	human silhouette [7], shape [8], edge [9]	video information	It is easy to obtain features, but it will be affected by the environment and background, resulting in errors in the extracted feature information.
	angle [10,11,12]	bone information	It is difficult to be affected by the environment, but the effect of feature extraction for complex actions is poor.
dynamic feature parameters	optical flow [13]	video information	When the target is large, it will get good results. When the target is small, it is very difficult to extract and track features.
	motion trajectories [14,15,16]	bone information	It is difficult as it is affected by the complexity of the moving scene, but there is noise interference to the extracted feature information.

**Table 2 sensors-22-07386-t002:** Kinect gets 32 joint points.

Number of Joint Points	Letter of Joint Points	The Name of the Joint Points
1	O	PELVIS
2	/	SPINE_NAVAL
3	/	SPINE_CHEST
4	/	NECK
5	Q	CLAVICLE_LEFT
6	D	SHOULDER_LEFT
7	E	ELBOW_LEFT
8	F	WRIST_LEFT
9	/	HAND_LEFT
10	/	HANDTIP_LEFT
11	/	THUMB_LEFT
12	P	CLAVICLE_RIGHT
13	A	SHOULDER_RIGHT
14	B	ELBOW_RIGHT
15	C	WRIST_RIGHT
16	/	HAND_RIGHT
17	/	HANDTIP_RIGHT
18	/	THUMB_RIGHT
19	H	HIP_LEFT
20	J	KNEE_LEFT
21	L	ANKLE_LEFT
22	/	FOOT_LEFT
23	G	HIP_RIGHT
24	I	KNEE_RIGHT
25	K	ANKLE_RIGHT
26	/	FOOT_RIGHT
27	/	HEAD
28	/	NOSE
29	/	EYE_LEFT
30	/	EAR_LEFT
31	/	EYE_RIGHT
32	/	EAR_RIGHT

**Table 3 sensors-22-07386-t003:** Joint point selection.

Method	Selected Action	Joint Point Selection
overall joint selection	all actions	A–F G–L
local joint selection	punch	A–C
	one-armed call for help	A–C
	arms for help	A–F
	trip forward	A, D, G, H, O, P, Q
	trip backwards	A, D, G, H, O, P, Q

**Table 4 sensors-22-07386-t004:** Overall action vector selection.

Method	Selected Action	Bone Feature Vector Selection
Overall “adjacent joint method”	all actions	$\vec{AB},\;\vec{BC},\vec{DE},\vec{EF},\vec{GI},\vec{IK},\vec{HJ},\;\vec{JL}$
Overall “pelvis divergence method”	all actions	$\vec{OA},\vec{OB},\vec{OC},\vec{OD},\vec{OE},\vec{OF},\vec{OG},$ $\vec{OH},\vec{OI},\vec{OJ},\vec{OK},\vec{OL}$

**Table 5 sensors-22-07386-t005:** Local action vector selection.

Method	Selected Action	Bone Feature Vector Selection
Local “adjacent joint method”	Punch	$\vec{AB},\vec{BC}$
	One-armed call for help	$\vec{AB},\vec{BC}$
	Arms for help	$\vec{AB},\vec{BC},\vec{DE},\vec{EF}$
	Trip forward	$\vec{PA},\vec{QD},\vec{OG},\vec{OH}$
	Trip backwards	$\vec{PA},\vec{QD},\vec{OG},\vec{OH}$
Local “pelvis divergence method”	Punch	$\vec{OA},\vec{OB},\vec{OC}$
	One-armed call for help	$\vec{OA},\vec{OB},\vec{OC}$
	Arms for help	$\vec{OA},\vec{OB},\vec{OC},\vec{OD},\vec{OE},\vec{OF}$
	Trip forward	$\vec{OA},\vec{OP},\vec{OQ},\vec{OD},\vec{OG},\vec{OH}$
	Trip backwards	$\vec{OA},\vec{OP},\vec{OQ},\vec{OD},\vec{OG},\vec{OH}$

**Table 6 sensors-22-07386-t006:** Group settings and numbers of experimental conditions.

Property of Angle	Model	Number
Sum of angle differences (A)	Overall “adjacent joint method”	A1
	Overall “pelvis divergence method”	A2
	Local “adjacent joint method”	A3
	Local “pelvis divergence method”	A4
Mean of angle difference (B)	Overall “adjacent joint method”	B1
	Overall “pelvis divergence method”	B2
	Local “adjacent joint method”	B3
	Local “pelvis divergence method”	B4
Maximum angle difference (C)	Overall “adjacent joint method”	C1
	Overall “pelvis divergence method”	C2
	Local “adjacent joint method”	C3
	Local “pelvis divergence method”	C4

**Table 7 sensors-22-07386-t007:** Overall “adjacent joint method” identification results.

Unsafe Behavior	Number of Actions	Number of Recognitions	Sum	Mean	Maximum
Punch	60	60	58.9%	60.2%	92.1%
One-armed call for help	60	60	71.6%	76.0%	85.9%
Arms for help	60	60	65.8%	72.0%	81.4%
Trip forward	60	60	85.4%	91.8%	80.6%
Trip backwards	60	60	82.1%	90.5%	96.5%

**Table 8 sensors-22-07386-t008:** Overall “pelvis divergence method” identification results.

Unsafe Behavior	Number of Actions	Number of Recognitions	Sum	Mean	Maximum
Punch	60	60	60.3%	60.3%	95.4%
One-armed call for help	60	60	75.3%	80.0%	96.6%
Arms for help	60	60	68.8%	67.1%	95.9%
Trip forward	60	60	87.9%	85.4%	88.6%
Trip backwards	60	60	87.2%	93.3%	95.0%

**Table 9 sensors-22-07386-t009:** Local “adjacent joint method” identification results.

Unsafe Behavior	Number of Actions	Number of Recognitions	Sum	Mean	Maximum
Punch	60	60	72.2%	80.4%	84.5%
One-armed call for help	60	60	65.3%	75.6%	79.6%
Arms for help	60	60	51.2%	68.8%	76.9%
Trip forward	60	60	69.3%	74.6%	79.1%
Trip backwards	60	60	73.2%	70.0%	75.1%

**Table 10 sensors-22-07386-t010:** Local “pelvis divergence method” identification results.

Unsafe Behavior	Number of Actions	Number of Recognitions	Sum	Mean	Maximum
Punch	60	60	79.2%	80.4%	86.9%
One-armed call for help	60	60	81.2%	75.2%	89.2%
Arms for help	60	60	65.7%	67.9%	85.5%
Trip forward	60	60	78.3%	80.2%	86.7%
Trip backwards	60	60	75.6%	81.7%	88.3%

**Table 11 sensors-22-07386-t011:** Test results using the improved identification method.

Behavior Type	Test Behavior	Test Times	Correct Judgment Times	Average Similarity	Early Warning
Safety behavior	Stand	10	10	65.2%	10 green 0 red
	Squat	10	10	43.7%	10 green 0 red
Unsafe behavior	Punch	10	9	86.9%	0 green 10 red
	One-armed call for help	10	9	89.2%	0 green 10 red
	Arms for help	10	9	85.5%	0 green 10 red
	Trip forward	10	10	86.7%	0 green 10 red
	Trip backwards	10	10	88.3%	0 green 10 red

**Table 12 sensors-22-07386-t012:** Test results using traditional identification methods.

Behavior Type	Test Behavior	Test Times	Correct Judgment Times	Average Similarity	Early Warning
Safety behavior	Stand	10	10	62.1%	10 green0 red
	Squat	10	10	40.5%	10 green 0 red
Unsafe behavior	Punch	10	0	60.2%	8 green 2 red
	One-armed call for help	10	6	76.0%	0 green 10 red
	Arms for help	10	5	72.0%	1 green 9 red
	Trip forward	10	10	91.8%	0 green 10 red
	Trip backwards	10	10	90.5%	0 green 10 red

## Data Availability

Not applicable.

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
