# Peer review of "An Improved Kinect Recognition Method for Identifying Unsafe Behaviors of Metro Passengers"

_sensors, 2022, doi:10.3390/s22197386_

Round 1
Reviewer 1 Report
The authors of the manuscript titled "An Improved Kinect Recognition Method for Identifying Unsafe Behaviors of Subway Passengers" conducted interesting and important work. After going through the work, I have following comments:
1. Clearly mentioned primary contribution of the manuscript in the article, preferably in introduction section. 2. There are several other works available which had performed similar work. Regression on a transfer learning model can be easily used to detect pelvis or other body parts and several works have performed similar work. Now further, the movements can be processed to identify the action they are performing. Based on the text in the manuscript, it is not yet clear as to why this work improves the state of the art methods. Please clarify the novelty and motivation of the work by stating the limitations of the state of the art which the presented work intends to solve. 3. The y-axis have not been labeled in many of the graphs shown in the paper. Example, fig. 10. Please fix these.Author Response
Please see the attachment.

Reviewer 2 Report
This paper presents an improved action (unsafe behaviors of subway passengers) recognition method using Kinect sensor. The authors identified five unsafe behaviors for recognition; collecting data and applying the DTW method detect the behavior. The presented work has major issues, notably:
1) The efficiency of proposed method is low, not good for action recognition as validated by the authors;
2) The proposed techniques lack of motivation in each proposal;
3) The technical writing needs significantly improvement; there are many typos (i.e., passages, check the reference list);
4) some figures with very low quality, i.e., Fig. 12;
5) lack of comparison with relevant methods.
Reviewer 3 Report
I have reviewed this paper with my best efforts and skills; the review comments are mentioned as below:
The Paper is interesting and well done and written with a good number of references.
The Introduction part should mention a detailed research design.
Round 2
Reviewer 2 Report
The authors have addressed my comments and the updated version has been improved. Thought I suggest to accept this work, but the authors should carefully check again the presented results and discussions
